# The Magnetic Technique—A Novel and Promising Method to Improve Axillary Staging Localisation from a Swedish Perspective

**DOI:** 10.3390/medicina59101727

**Published:** 2023-09-27

**Authors:** Fredrik Wärnberg, Christine Obondo, Kian Chin

**Affiliations:** 1Institute of Clinical Sciences, Sahlgrenska Academy, University of Gothenburg, 41345 Gothenburg, Sweden; 2Department of Surgery, Sahlgrenska University Hospital, Region Västra Götaland, 41345 Gothenburg, Sweden; 3Department of Surgery, Södersjukhuset, 11883 Stockholm, Sweden

**Keywords:** breast cancer, magnetic tracer, magnetic clip

## Abstract

The magnetic technique using superparamagnetic nanoparticles of iron oxide has been well established for sentinel lymph node detection. Its main advantage is in the context of logistics, with the possibility to inject several weeks before surgery and the possibility to give access to sentinel lymph node biopsy for women worldwide in places without nuclear medicine facilities. We have not yet seen the full potential of this technique, and new implications have been developed for breast tumour localisation with paramagnetic clips and axillary staging after neoadjuvant chemotherapy using paramagnetic clips inserted in lymph node metastases before chemotherapy. In this report, we have presented our experience of the magnetic technique starting in 2014, and we have highlighted our current and future research directions.

## 1. Introduction

In recent years, the superparamagnetic iron oxide (SPIO) nanoparticles tracer and magnetic seed localisation (Magseed^®^, Endomagnetics, Cambridge, UK) have emerged as novel techniques for sentinel lymph node biopsy (SLNB) and for detecting non-palpable breast tumours in breast cancer surgery [1,2]. Our research group first participated in a SPIO study [3] in 2014. We have subsequently established SPIO as our standard tracer for SLN detection, replacing both technetium and blue dye. Concurrently, we have conducted several clinical studies, aiming to improve the SPIO technique, to find new implications for the tracer and to explore its possible side effects. In this overview, we have presented our experience, implementing the magnetic technique in Sweden, and we have discussed recent developments and future directions. 

The technique of sentinel lymph node biopsy revolutionised axillary staging in breast cancer management in the last 25 years. Previously, in clinically node-negative patients, all level I and II lymph nodes were removed during the axillary staging procedure. With the transition to routine SLNB today, there has been a significant decrease in arm and shoulder morbidity compared to staging with axillary lymph node dissection (ALND). SLNB has reduced operating times and lessened the pathology resources required to analyse specimens. During the first 20 years of using this technique, SLN detection with dual tracers, namely Technetium^99^ (Tc^99^) and blue dye, was the gold standard. The main disadvantages still include the short half-life of the radioisotope, the low availability of nuclear medicine facilities worldwide, and the rare anaphylactic reactions associated with blue dye. The drawbacks of this magnetic technique have been skin staining and the presence of MRI artefacts.

SPIO has been validated and well established as a SLN tracer. More than 200,000 women in over 40 countries have had SLNB using the magnetic technique. It is non-inferior to the dual technique for SLN detection [4,5], and this has been demonstrated in several studies, including the Nordic trial in which five Swedish hospitals participated [3]. In this trial, no allergic reactions were reported. In our experience, it was easy to implement; it has now been routinely used for more than five years across several hospitals in Sweden, and no nuclear medicine facilities are required. Indeed, our experience is transferrable to most countries, and it can help simplify the logistics around planning breast cancer surgery.

## 2. Sentinel Lymph Node Tracer

SPIO contains nanoparticles which measure approximately 50 nm. Like other tracers, these particles drain via the lymphatic channels into the SLN, where they are engulfed by macrophages. The SLN can then be detected using a hand-held magnetometer probe (SentiMag^®^, Endomagnetics, Cambridge, UK) in a similar manner as with the gamma probe and Tc^99^. Intraoperatively, SPIO is seen as a brown stain within the SLNs, which aids in the detection process. Unlike blue dye, SPIO does not stain the lymph vessels, but its staining properties are nonetheless as distinct as the blue colour of the dye. 

Our research group has studied the effectiveness of the magnetic technique compared to the dual technique and explored novel applications of its use in breast and melanoma surgery. We have demonstrated that its detection rate is comparable to the dual technique in our hands [3]. In the initial studies, larger volumes of SPIO were used. We first started with 2 mL of SPIO (Sienna+) diluted with 3 mL of saline and injected this in the retro-areolar region. This was given at least 20 min before surgery and was followed by a five minute breast massage. The resultant detection rate using this technique was non-inferior to the dual technique. With these volumes, a brownish skin stain developed at the injection site in approximately 65% of women undergoing breast-conserving surgery [3], and the mean size of the stainings was 25 cm^2^.

Since the above early studies, the SPIO injection technique has been adapted. In our institution today, 1 mL of undiluted SPIO (Magtrace^®^, Endomagnetics, Cambridge, UK) is injected deeper into the breast, adjacent to the tumour. The injection is mainly given by the surgeon at the appointment when the surgery is planned. The timing is approximately two weeks before surgery. In patients with impalpable lesions, the injection is given freehand close to the tumour position within the breast. In lesions close to the axilla, the injection is given medial to the lesion away from the axilla. This makes detection with the magnetometer probe easier. In most of our research projects, the SPIO injection is given by the research nurses to prompt standardisation. In one of the projects, however, the injection was given by the radiologists concurrently with Magseed^®^ insertion. This was because the SPIO injection site was important in the given project, and it was therefore performed under ultrasound guidance [6].

Due to smaller volumes and deeper injections, skin staining is now much more seldom seen. The size of any staining observed nowadays is significantly smaller than in the earlier trials. After a deep peri-tumoural injection of 1.0 mL of SPIO, 11.4% of women having BCS had a staining with a mean size of 8.5 cm^2^ [7]. The duration of residual staining in our experience is up to a range from two to three years, and it fully resolves in 90.6% of the women at three years. This time frame of resolution is similar to that of the blue dye staining [7]. In our experience, most patients seldom considered staining a cosmetic problem. In patient-reported outcomes, 88% of the women studied described it as either “no” or as a “minor cosmetic problem”. Nevertheless, we continue to explore further ways of reducing SPIO-related side effects, and we are currently conducting a study in which an ultra-low dose of Magtrace^®^ (i.e., 0.1 mL intradermally) is being evaluated. A pilot study with 50 patients undergoing breast-conserving surgery has already been successful in SLN detection in all the case subjects. We have now launched a larger, multicentre trial of the ultra-low dose and are evaluating skin staining and the presence of MRI artefacts in the pilot cohort.

The timing of the injection has also been a subject of investigation. In the SentiDose trial, injecting SPIO up to seven days before the day of surgery increased the SN detection rate (100% vs. 97.5%, *p*  =  0.11) compared to injecting SPIO on the day of surgery [7]. However, this difference was not statistically significant. In this trial, a mean of 0.3 more SLNs were retrieved per patient when the injection was given several days before surgery. In the SentiNot trial discussed below [8], we demonstrated that SPIO can be successfully injected four weeks before surgery. In practice today, SPIO is injected at the preoperative outpatient clinic and the operation is completed within four weeks of seeing the patient. We have found that the logistics around planning surgery become easier when compared with Tc^99^, which has a half-life of approximately six hours. 

SentiNot [8] was a prospective feasibility study in which the SLN was marked with SPIO during the primary surgery weeks before an eventually delayed SLNB, allowing patients with ductal breast carcinoma in situ (DCIS) to avoid unnecessary axillary surgery. These patients had high-risk DCIS and were undergoing BCS or mastectomy for low-risk disease. SPIO was injected preoperatively, but the SLN was not removed. If DCIS was upgraded to invasive breast cancer, SLNB was performed as a delayed SLNB (d-SLNB) within four to five weeks after the primary surgery. The d-SLNB was performed with Tc^99^ and blue dye as well, according to the protocol and according to routine clinical practice. A total of 254 patients were recruited, and 55 d-SLNBs were performed using SPIO and the dual technique as the control. The results showed that SPIO persisted in the SLN for four weeks, and that SLNB could be avoided in 80% of women with DCIS when there was no upgrade to invasive cancer on the final histopathology assessment. This resulted in a reduction of the risk of arm morbidity and led to an overall cost saving [8]. Another finding from the SentiNot trial was that compared to the dual technique, more than half of the SLNs retrieved via d-SLNB were not concordant between the two techniques, SPIO before surgery versus the dual technique after primary surgery. It appears that more lymph nodes are harvested at d-SLNB, and these may not all be true SLNs. However, continuing with an upfront SLNB leads to many unnecessary axillary procedures in women with DCIS. Further research is required in this area, and hence our SentiNot 2.0 trial was planned and is underway. This is an international, multicentre study aiming to recruit 2000 patients in order to reach about 500 d-SLNBs (NCT04722692). 

We continue to study how long SPIO persists within SLNs and its role in the neoadjuvant setting. In the Swedish SentiNeo pilot study (discussed below), SPIO was injected into the breast prior to initiating neoadjuvant chemotherapy. The timing was several months before planned axillary surgery. A total of 80 patients have been recruited and the trial is ongoing. Therefore, there are no preliminary data available as of yet, but early indications show that SPIO stay in the SLNs for much longer than four weeks.

During breast cancer investigations, SPIO has not been found to interfere with mammogram readings nor ultrasound performance. The tracer, however, affects magnetic resonance images (MRIs), which has raised concerns about using it in breast-conserving surgery. A SPIO injection given into the breast has been shown to induce black void artefacts on MR images that last several years [9]. Hence, these long-term MRI effects require further evaluation. Our hypothesis is that through utilising a lower volume of SPIO and injecting it peri-tumourally, most of the SPIO will be resected within the tumour specimen and this will probably minimise artefacts. In the above-mentioned pilot trial using ultra-low SPIO intradermally, we hope that the SLN detection rates will not only be good, but that the impact on postoperative MRI will be negligible with an intradermal dose of 0.1 mL of SPIO. Additionally, we expect the ultra-low dose to be more compatible with the Magseed^®^ localisation of non-palpable breast lesions. Theoretically, the ultra-low dose can be injected at the areolar border, and it can therefore be separated from the magnetic clip in most cases. This would reduce cross signalling between SPIO and the magnetic clip when using the SentiMag probe. However, this has not yet been proven. Indeed, the optimal method for using SPIO for SLN detection with regard to volume, timing, injection site, and MRI optimisation has yet to be determined.

As SPIO is visualised via MRI, it can be used as a SLN localiser [10]. SPIO is injected interstitially or intradermally into the breast and MRI of the axillae is subsequently performed. The localised lymph nodes become visible after a couple of minutes. However, the ideal timing of the injection before MRI still needs to be established. This technique, however, appears to show promise in melanoma surgery. With melanomas, the SLN basin may be located close to the primary tumour, or it may lie further afield in the axillae and groins. We have already demonstrated in limb melanomas that the SLNs can be located via MRI [11]. The primary lesion was firstly excised according to standard clinical routine. Secondly, 15 consecutive patients requiring further re-excision and SLNB were recruited into a pilot study. An ultra-low dose of SPIO was intradermally injected in the four quadrants around the scar. The SLN was then detected successfully in all 15 patients aided via MRI. Deriving from these findings, a larger multinational study has been conceived and it is underway. 

Changes to the lymphatic network are known to occur following prior axillary surgery for breast cancer. It has been noted that the new SLN may lie in the contralateral axilla or even in the internal mammary chain. At the Sahlgrenska University Hospital, Gothenburg, Sweden, we are currently using MRI to map SLNs in patients with ipsilateral local breast cancer recurrences requiring repeat axillary surgery. These patients had previously been staged through either SLNB or axillary node dissection. Our clinical routine has been to perform repeat SNB with Tc^99^ and to map the node preoperatively with lymphoscintigraphy. Presently, we are also injecting 1–2 mL of SPIO at the recurrence site in 20 patients and visualising the draining nodes on MRIs. We will evaluate and compare the use of both tracers in the setting of repeat SLNB. Recently, we presented very early results at the Swedish Surgical week based on the first 15 patients. Using SPIO with MRIs of the axillae and SentiMag probe identified at least one SLN in more patients (60%) compared to Tc^99^ with lymph scintigraphy and the gamma probe. 

## 3. Magnetic Clip

In addition to SPIO, a paramagnetic steel clip (Magseed^®^) has been developed for tumour localisation for breast-conserving surgery. The clip is placed, guided either through ultrasound or stereotactically, and it is localised with the same SentiMag probe as the SPIO in the SLNs. In many countries, the magnetic seed has replaced the commonly used steel wire for breast tumour localisation [12,13]. As the magnetic clip can be placed at any time before surgery, it makes the logistics easier for surgical planning. This contrasts with the steel wire localisation, which is usually placed on the day of surgery, or occasionally on the day prior to surgery. The CE mark of the Magseed^®^ also allows for the clipping of tumours for localisation before neoadjuvant chemotherapy, several months before surgery. Today, we use a metallic clip for tumour localisation before NACT followed by a steel wire localisation of the clip if the tumour is not palpable after NACT. This secondary localisation of the clip is not necessary if Magseed^®^ is used initially. The experience in the UK [14] suggested that Magseed^®^ is a feasible method of breast localisation and may provide additional benefits over steel wire localisation for surgical scheduling and improved patient flow. In the study by Morgan et al. [15], they listed difficult percutaneous localisation, interference with surgical instruments, and the need for probe recalibration as potential drawbacks of this technique. On the other hand, the learning curve was short for surgeons experienced in other localisation techniques.

Magseed^®^ is also used to clip metastatic axillary lymph nodes before neoadjuvant chemotherapy (NACT) in clinically node-positive patients (cN+) [14]. This ensures that the correct lymph node/nodes are removed during surgery after NACT, and allows for the tumour response to NACT in the lymph nodes to be assessed. If the tumour response is good, the metastatic nodes could sometimes be indistinguishable from the other nodes if no clip is inserted preoperatively. This technique is usually carried out synchronously with a SLNB, which is then subsequently termed as targeted axillary dissection (TAD). TAD reduces the risk of false-negative staging in the axilla after NACT, and in the context of a complete pathological response in the axilla, patients require no further surgery in the form of ALND. If ordinary non-magnetic metallic clips are used to mark the metastatic nodes prior to NACT, the clip has to be localised after NACT with a steel wire before surgery. Using the Magseed^®^ saves the patient from having a secondary localisation procedure and saves resources.

If a tumour response is measured via MRI during NACT, a magnetic clip within an axillary lymph node may affect the interpretation of the MRI. In the ongoing SentiNeo study (mentioned above), SPIO is injected into the breast prior to NACT, and in node-positive patients a Magseed^®^ is also placed in the metastatic lymph node. Evaluation of the tumour response via MRI is one exclusion criterion of the study. In our hospital in Gothenburg, we usually follow tumour responses with mammography and ultrasound, and MRI is indicated for follow up in only 10% of cases. This follow-up method is predetermined at the multidisciplinary team conference prior to commencing preoperative treatment. In the cases where the tumour response is followed by MRI, a magnetic clip is still inserted for index node localisation in the axilla, but SPIO for SLN detection is injected after NACT out of the study. 

The main aim of the SentiNeo study is to verify the feasibility of injecting SPIO three to four months before surgery, but also to find out if chemotherapy affects the lymphatic drainage. To study this, we compared the nodes that are detected via the magnetic technique (SPIO injected before NACT) with the nodes that are identified with Tc^99^ injected after NACT to see if the nodes are concordant. The results from the pilot study will be presented in the autumn of 2023. Another aim of the SentiNeo pilot study was to evaluate if it is surgically feasible to remove a metastatic lymph node marked with a magnetic clip and to then perform SLNB with SPIO in the same axilla. We used the same probe to localise these nodes, and to date we have not encountered any problems surgically when distinguishing the index metastatic node from the SLNs. 

From a practical perspective, the same method used for breast tumour localisation and SLN detection in the same patient could potentially further streamline planning logistics for surgery. On that basis, we have now successfully conducted a pilot study combining Magseed for tumour localisation and SPIO for SLN detection [6]. A total of 32 patients were recruited and were able to achieve tumour-free resection margins, with a minimum of one SLN detected in all 32 patients. Both the radiologists and surgeons reported that the procedure was easy to carry out, and more specifically that the technique facilitated the planning of both the localisation and surgery. Following on from the pilot study, a randomised study including more than 400 patients (MagTotal trial/ISRCTN11914537) comparing wire and Magseed tumour localisation, in combination with SPIO for SLN detection, has now been conducted at three Swedish hospital sites, namely Uppsala Academic Hospital, Västerås Hospital, and Sahlgrenska University Hospital. In the MagTotal randomised study, the Magseed^®^ was placed anterior to the breast lesion and 1.0 mL of SPIO was injected by the radiologist either posterior or adjacent to the lesion at the same time as the Magseed^®^ insertion. The idea was to bracket the lesion in between the Magseed and SPIO. The preliminary results have been presented, and an interim analysis of the first 207 patients of the study was published in PhD Hersi’s thesis [6], showing that there was no statistically significant difference in reoperation rates due to inadequate margins between Magseed^®^ and wire localisation. Additionally, no significant difference in SLN detection rates was seen. The full report of the MagTotal study has been submitted for publication.

## 4. Future Aspects of the Magnetic Technique

As mentioned above, the optimal volume of the injected SPIO, injection site, and timing of the injection has yet to be fully established for the magnetic technique. Improvements will have implications for the SLN detection rate, planning for surgery, staining of the skin, and MRI artefacts. The use of the combination of a magnetic tracer and magnetic clips for breast lesion localisation will undoubtedly see improvements with further technical refinements in the next generation of detection probes. Today, an injection of SPIO close to a magnetic clip could make the breast tumour localisation difficult, as the signal from a large volume of SPIO can mask the signal from the clip. In our MagTotal study, using the combination of Magseed^®^ and SPIO, we chose to insert the clip and tracer close together in order to be able to remove most of the tracer during surgery and to minimise postoperative MRI artefacts. On the other hand, distancing the tracer injection from the magnetic clip could make tumour localisation using the probe easier, with less signal disturbances from the SPIO in the proximity. An ultra-low dose of SPIO, with comparable SLN detection rates as 1.0 mL of Magtrace^®^, injected intradermally, away from the tumour site could also reduce interferences when using the probe, as well as minimising MRI artefacts. 

Another possible technical development is a next-generation magnetic clip with a much smaller degree of MRI artefacts. Preliminary pre-clinical data presented at the San Antonio Breast Cancer Conference 2022 by Thiruchelvam et al. [16] indicated that there could be beneficial implications for the use of magnetic clips in patients undergoing NACT, both for the tumour and index node metastasis localisation.

Currently, there have been compatibilities issues using metallic surgical instruments during an operation where a magnetic approach has been adopted. Specifically, any metallic surgical instruments must be removed from the operating area when the magnetic detector probe is used in the wound; furthermore, the system needs calibrating almost every time the probe is handled. Although plastic surgical instruments have been developed, they have generally been difficult to handle and ineffective, such that many surgeons preferred not to use them. However, there have been improvements in the plastic instruments, and the recalibration of this system will hopefully be much easier with the future generation of magnetic probes. Although calibration of the probe takes only a few seconds, compared to the gamma probe, this procedure is more time consuming and cumbersome for surgeons who are new to it.

The issue of determining an optimal dose has remained largely unresolved. In the MagMen study, doses of SPIO as small as 0.02 mL (net volume 0.2 mL) were injected intradermally for limb melanomas, with good results in SLN detection [11]. The volume used in our last breast cancer study (0.1 mL) was 50 times lower than the 2 mL of SPIO + 3 mL of saline volume used in the first trials [6]. We do not know if it is the volume, in itself, or the iron oxide nanoparticle content that matters for a successful SLN detection. It is likely that both factors are important, but this remains to be determined. A coincidental finding using the very small doses/volumes of Magtrace^®^ in the MagMen study [11] was that metastases in the SLNs could be visualised on the axillary MR images in some cases. These findings are preliminary, but, nevertheless, the concept of SPIO being used as a metastatic tracer is something worth exploring in the future. Similar findings have also been reported in a Japanese study [10].

SPIO, as a magnetic tracer for sentinel lymph node detection, is nowadays being explored for many other diagnoses than breast cancer and malignant melanoma. This widespread adoption of SPIO will only increase the potential for new implications and technique development in the future. Surgeons involved in head and neck surgery, gynaecology, and colorectal surgery will add new ideas and further improvements. Additionally, radiologists will undoubtedly contribute to better uses of MRI for SLN localisation, staging, and optimisation of combining SPIO and magnetic clips.

## 5. Conclusions

The magnetic technique using SPIO is now a well-established method for SLN detection. The main advantage is the possibility to facilitate logistics and surgery planning. Another advantage is its ready availability with no need for nuclear medicine facilities, thereby allowing worldwide breast cancer patients easy access to SLNB. The magnetic technique is novel, and we have yet to see its full potential. The use of magnetic clips for tumour localisation during breast-conserving surgery functions well, but there is yet no good scientific data available. The accuracy of the combined technique of the magnetic tracer and the magnetic clip for breast tumour localisation and SLN detection will develop further with the advent of a next-generation magnetic detecting probe, where it is possible to differentiate signals originating from a magnetic clip and Magtrace^®^. The marking of lymph node metastases before neoadjuvant chemotherapy seems to be a feasible method for targeted axillary dissection, assisted using SPIO for simultaneous SLN detection. 

So far, the side-effects of skin staining and postoperative MRI artefacts have raised concerns regarding if patients require future MRI investigations. However, many studies focusing on reducing postoperative MRI artefacts and skin staining are underway. It appears that reducing the volume of SPIO and deeper injections in the breast may be helpful. Early data on a magnetic seed with less effect on MRI for tumour and lymph node localisation [16] are also promising for patients undergoing neoadjuvant chemotherapy where their tumour responses have to be evaluated with MRI.

## Data Availability

Data for each individual study is available on request.

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
