# Peer review of "The Magnetic Technique—A Novel and Promising Method to Improve Axillary Staging Localisation from a Swedish Perspective"

_medicina, 2023, doi:10.3390/medicina59101727_

Round 1

Reviewer 1 Report

This is my review report for manuscript: The magnetic technique – A novel and promising method to improve axillary staging localisation from a Swedish perspective. Paper is about current. The paper discusses current knowledge and the application of magnetic technique in axillary staging in breast cancer, as well as potential future aspects of this novel technique. I believe that this topic is very interesting, particularly considering its applicability in hospitals where nuclear medicine services are not available. Some minor corrections are needed in the paper to make it suitable for publication.

Citing the references is not appropriate; the first cited paper is numbered 2, and the citations should start with the first reference on the list. I believe this is simply a technical oversight by the authors. In-text citation should follow the order of references on the list. There's no need to write "ref" in parentheses, only the reference number you are citing from the list.

In the second paragraph of the "Sentinel Node Tracer" section, references are missing; you wrote (ref the papers) and (ref)...

In the second paragraph of the introduction, it might be better to say: all level I and II lymph nodes were removed instead of 10 to 15.

In the same paragraph towards the end of the page, it might be better to say "blue dye" since most hospitals still use methylene blue rather than patent blue

At the end of the first paragraph in the "Magnetic Clip" section, you mention the study by Morgan et al., but you haven't provided the reference number, and I cannot find it in the list of references

All references must be within the last three years in order for the paper to be categorized as a perspective article

Minor editing of English language required

Author Response

Reviewer 1

A novel and promising method to improve axillary staging localisation from a Swedish perspective. Paper is about current. The paper discusses current knowledge and the application of magnetic technique in axillary staging in breast cancer, as well as potential future aspects of this novel technique. I believe that this topic is very interesting, particularly considering its applicability in hospitals where nuclear medicine services are not available.

Some minor corrections are needed in the paper to make it suitable for publication.

Citing the references is not appropriate; the first cited paper is numbered 2, and the citations should start with the first reference on the list. I believe this is simply a technical oversight by the authors. In-text citation should follow the order of references on the list. There's no need to write "ref" in parentheses, only the reference number you are citing from the list.                                                        --Thank you for this comment. We have looked through the references and made the changes required.

In the second paragraph of the "Sentinel Node Tracer" section, references are missing; you wrote (ref the papers) and (ref).                                                            -Thank you again. This has been sorted.

In the second paragraph of the introduction, it might be better to say: all level I and II lymph nodes were removed instead of 10 to 15.                                             -We have changed this accordingly.

In the same paragraph towards the end of the page, it might be better to say "blue dye" since most hospitals still use methylene blue rather than patent blue.

-This now changed.

At the end of the first paragraph in the "Magnetic Clip" section, you mention the study by Morgan et al., but you haven't provided the reference number, and I cannot find it in the list of references,

-Thank you, we missed the reference. It has been included (17).

All references must be within the last three years in order for the paper to be categorized as a perspective article.                                                                          -As we refer to our own experience of the magnetic technique we think it is appropriate to reference all the papers published throughout the period described.

Fredrik Wärnberg

Reviewer 2 Report

The authors present an impressive up-to-date review on magnetic tracers and markers for breast cancer patients. I recommend to accept the article after minor revisions. 

1. Introduction - "the anaphylactic reactions associated with blue dye." It should be mentioned, that the incidence of anaphylactic reactions is quite low. 

2. Introduction - I recommend to mention the disadvantages of SPIO in the Introduction because you are mentioning only the advantages. 

3. Page 2 - "(ref the papers)" References are missing?

4. Page 3 - "The sizes of any staining seen is significantly smaller than in the earlier trials (ref 4)." Please add the specific numbers from the study to compare sizes. 

5. Page 4-5 - Please, do you have any preliminary results about re-SLNB, especially the success rate? 

6. I recommend to discuss the interference between SPIO and Magseed in the breast. 

7. References - Reference number 18 is not in the right formatting, also I found only 19 references for the review article really low - e.g. I recommend adding more citations to the Introduction, there are many of information without citations. 

Author Response

Reviewer 2

The authors present an impressive up-to-date review on magnetic tracers and markers for breast cancer patients. I recommend to accept the article after minor revisions. 

  1. Introduction - "the anaphylactic reactions associated with blue dye." It should be mentioned, that the incidence of anaphylactic reactions is quite low.

-We have added “rare” to the sentence.

  1. Introduction - I recommend to mention the disadvantages of SPIO in the Introduction because you are mentioning only the advantages. 

-This has been added to the second paragraph in the “Introduction”.

  1. Page 2 - "(ref the papers)" References are missing?

-See above. All references have been revised.

  1. Page 3 - "The sizes of any staining seen is significantly smaller than in the earlier trials (ref 4)." Please add the specific numbers from the study to compare sizes. 

-This information is added.

  1. Page 4-5 - Please, do you have any preliminary results about re-SLNB, especially the success rate? 

 -We cannot give the full details as we have submitted an abstract to the San Antonio Breast Cancer Conference in December. However, we have added very preliminary data.

  1. I recommend to discuss the interference between SPIO and Magseed in the breast. 

-We have extended the first paragraph in the section “Future aspects…..” where we discuss the combined technique.

  1. References - Reference number 18 is not in the right formatting, also I found only 19 references for the review article really low - e.g. I recommend adding more citations to the Introduction, there are many of information without citations.

-We have tried to re-format the reference no numbered as reference 19. It is the thesis of Abdi Hersi 2021. We also added two references to the Introduction. 

Fredrik Wärnberg